*Environmental Data Science* (2026), 5: e6, 1–18

CAMBRIDGE
UNIVERSITY PRESS

**APPLICATION PAPER**

# Skillful subseasonal Indian Ocean marine heatwave forecasts using a neural network

Lucas Howard[1] , Aneesh C. Subramanian[1], Jithendra Raju Nadimpalli[2], Donata Giglio[1] and Ibrahim Hoteit[2]

[1]Atmospheric and Oceanic Science, University of Colorado Boulder, USA
[2]Physical Science and Engineering, King Abdullah University of Science and Technology, Saudi Arabia
**Corresponding author:** Lucas Howard; Email: lucas.howard@colorado.edu

**Keywords:** marine heatwave; machine learning; S2S; U-Net; Indian Ocean

**Abstract**

Marine heat waves (MHWs) are prolonged periods of elevated ocean temperatures that can devastate marine ecosystems, fisheries, and coastal communities. Skillfully predicting these events with sufficient lead time is crucial for mitigating their adverse effects. This study presents a probabilistic subseasonal MHW forecast tool using a U-Net-based neural network architecture, with a focus on the Northern Indian Ocean and the Arabian Sea. The model was trained using sea surface temperature and sea surface height reanalysis data. The U-Net-based forecast tool demonstrated significant predictive skill up to 10 weeks in advance across various deterministic and probabilistic skill metrics. The model outperformed persistence and climatology-based benchmarks, especially in the tropical warm pool. Future applications of explainable artificial intelligence (XAI) methods have the potential to identify the sources of predictive skill, inform understanding of underlying dynamics, and improve dynamic subseasonal to seasonal forecast models.

**Impact Statements**

Periods of unusually warm ocean surface temperatures, known as marine heatwaves, can have profound impacts on marine ecosystems and coastal communities. These events can also impact the weather through the interactions of the ocean with the atmosphere. The ability to predict marine heatwaves in advance is crucial in enabling local stakeholders to prepare. We trained a common machine learning method to predict ocean surface temperatures in the Indian Ocean at lead times of up to 10 weeks. Its performance is competitive with existing forecast methods.

## 1. Introduction

Marine heatwaves (MHWs) are periods of anomalous and extreme warm ocean temperatures that last from days to months. Depending on severity, duration, and location, MHWs can have significant impacts on wildlife, ecosystems, fisheries, and aquaculture (Smale et al., 2019; Oliver et al., 2021; Guo et al., 2022). MHWs have become more frequent and intense over the last century (Oliver et al., 2018), and this

 This research article was awarded Open Data and Open Materials badges for transparent practices. See the Data Availability Statement for details.

trend is expected to continue under projected climate change scenarios (Frölicher et al., 2018; Deser et al., 2024). As a result, significant efforts have been made to improve understanding of the dynamics of MHWs and enhance the skill of their forecasts (Holbrook et al., 2020).

MHWs are driven by a variety of physical processes operating across multiple time scales. Atmospheric forcing, planetary waves, variability in ocean circulation patterns, and teleconnections with climate modes, such as the El Niño Southern Oscillation, all influence ocean temperatures and the incidence of MHWs (Holbrook et al., 2019; Amaya et al., 2020; Oliver et al., 2021; Jacox et al., 2022; Hamdeno et al., 2024). Depending on the physical mechanism driving an MHW event, the predictability limits can range from weeks to years (Holbrook et al., 2020; Meehl et al., 2021).

Identifying forecasts of opportunity, where low-frequency dynamical modes offer the potential for relatively high predictive skill, is crucial for improving Earth system forecasts on subseasonal to seasonal (S2S) timescales (Albers and Newman, 2019; Lang et al., 2020; Mariotti et al., 2020). Ocean forecasts should similarly leverage these opportunities, where predictable and low-variability drivers enable high predictive skill in specific scenarios. Furthermore, since the coupling between the ocean and atmosphere is a significant source of atmospheric S2S predictability (Meehl et al., 2021), skillful S2S sea surface temperature (SST) forecasts offer the potential to improve the quality of atmospheric forecasts.

Dynamical forecast models have shown skill in S2S predictions of MHW events (Doi et al., 2013; Benthuysen et al., 2021; Spillman and Smith, 2021; Jacox et al., 2022; Mogen et al., 2023), indicating the existence of drivers with characteristic timescales on the order of S2S lead times. The relative computational expense of dynamical models, combined with the increasing volume of high-quality observational and reanalysis data, has spurred the development of machine learning (ML)-based forecast tools for MHW and SST prediction (Taylor and Feng, 2022; Yao et al., 2023; Davenport et al., 2024; Sun et al., 2024; Xie et al., 2024; Bachèlery et al., 2025). Many of these forecast tools, both dynamical and statistical, have either targeted shorter-term forecasts using daily mean temperature (Benthuysen et al., 2021; Sun et al., 2024; Xie et al., 2024) or longer-term forecasts using monthly mean temperature (Jacox et al., 2022; Taylor and Feng, 2022; Mogen et al., 2023; Bachèlery et al., 2025). However, most ML approaches to date have generated deterministic forecasts (e.g., Sun et al., 2024; Bachèlery et al., 2025), which may be less useful to decision-makers than probabilistic or risk-based forecasts.

Most of the ML applications to this problem have used neural networks (NNs), which are a subset of ML methods in which data are passed between one or more layers of connected nodes before a nonlinear activation function is applied. The weights of the connections are adjusted during training to minimize a loss function across a training dataset, typically a subset of available data, with the rest withheld for testing and/or validation. Convolutional NNs (CNNs) utilize layers of convolutional kernels and are particularly well-suited to spatial data. U-Nets, a subset of CNNs, are a deep learning NN architecture in which both large-scale and small-scale features are retained (Ronneberger et al., 2015). U-Nets have been successfully used in several earth science applications, including precipitation forecasting, forecast bias correction, and processing of satellite observations (Chapman et al., 2022; Nataraja et al., 2022; Taylor et al., 2022; Faijaroenmongkol et al., 2023). U-Nets have also been successfully used for the prediction of SST and MHWs (Taylor and Feng, 2022; Xie et al., 2024).

The Indian Ocean and Arabian Sea border regions have some of the highest population densities in the world. They account for a large portion of a variety of fisheries, and their dynamics have substantial impacts on air temperature and precipitation patterns over land (Vialard et al., 2012; Roxy et al., 2015). The ocean dynamics in this region primarily impact the surrounding land masses through the Indian Summer Monsoon, a seasonal cycle of precipitation driven by temperature gradients between the land and sea surfaces. The Indian Ocean also impacts more distant regions via teleconnections, such as the Madden-Julian Oscillation (Madden and Julian, 1971) and the Indian Ocean Dipole (Saji et al., 1999). Additionally, temperatures in the northern Indian Ocean have increased more than in other bodies of water as the climate has warmed, possibly amplifying these effects (Roxy et al., 2014; Sharma et al., 2023).

Planetary waves have been shown to drive MHW events in the Indian Ocean (Zhang et al., 2021), providing potential opportunities for forecasts. Using an ML-based approach, Bachèlery et al. (2025)

show that these planetary waves are sources of forecast skill for Atlantic Niño events on seasonal timescales. Regional MHW forecast tools (using both dynamical and ML approaches) have been generated for other bodies of water, including the Pacific Ocean (Spillman and Smith, 2021; Yao et al., 2023), Mediterranean Sea (Bonino et al., 2024), and South China Sea (Xie et al., 2024). However, comparatively little progress has been made in developing region-specific MHW forecast products for the Northern Indian Ocean and the Arabian Sea, despite their significance, particularly when compared with the Pacific and Atlantic basins for which multiple regional-scale operational forecast products are available.

To address this gap, this work presents a probabilistic ML-based forecast tool covering the northern Indian Ocean for S2S MHW forecasts. It forecasts SST at weekly time steps and up to 10 weeks of forecast lead time. The rest of the article is organized as follows: Section 2 presents the datasets used and the methods employed. Section 3 presents the performance of the ML forecast tool. Section 4 discusses exciting aspects of the work in more detail and potential future work, followed by Section 5, which presents the conclusion.

## 2. Methods

### 2.1. Data

The GLORYS12v1 ocean reanalysis dataset (Lellouche et al., 2021) was used to generate the training data for the ML forecast model. Reanalyses are commonly used for training ML forecast tools (Ham et al., 2019; Liu et al., 2021; Kim et al., 2022; Taylor and Feng, 2022; Kochkov et al., 2024; Cui et al., 2025). For our application, where planetary waves are the hypothesized sources of predictability, reanalysis is advantageous because it provides dynamically consistent SST–SSH linkages and avoids gaps present in purely observational data. SST was represented by the potential temperature at the uppermost layer (at a depth of 0.5 m). Daily mean values of SST and sea surface height (SSH) with a grid spacing of $\left(\frac{1}{12}\right)^{\circ}$, or ~8 km at the equator, were obtained for the period spanning from January 1, 1993, to June 30, 2021. The combination of SST and SSH should enable the detection of sources of predictability, such as planetary waves and ocean heat content anomalies. These variables were then temporally averaged to weekly mean values. Sources of predictability at S2S timescales will not capture daily variability; downsampling to a weekly frequency will dampen some of this noise and increase the likelihood of capturing an S2S signal.

The data were also downsampled spatially to speed model training. SST and SSH were interpolated onto a coarser regional grid covering the western Indian Ocean and Arabian Sea. The regional grid spans a latitude range from 12°S to 35°N and a longitude range from 30°E to 85°E, with 256 grid points in both the latitudinal and longitudinal directions. The study domain is shown in Figure 1. The downsampled grid has a spacing of 0.18° in the meridional direction and 0.21° in the zonal direction. While this creates variations in grid spacing, the approach is computationally convenient, the distortion is not large, and the resolution remains sufficient to capture hypothesized sources of predictability (equatorial waves). MHWs and equatorial waves both have characteristic length scales on the order of tens to hundreds of kilometers. As the downsampled grid spacing is finer than needed to resolve either MHWs or their precursors, constructing the ML model to use the original dataset is not needed.

Long-term trends and the seasonal cycle were removed from both variables before using them in the ML training process. First, a linear trend was fitted to the entire weekly mean time series and subtracted to generate anomalies. The seasonal cycle was then calculated by averaging each calendar week across the dataset and removed by subtracting it from the detrended anomalies. The final dataset exhibits no long-term temporal trend in SSH or SST, and no seasonal cycle is observed. Neither of these is of interest in generating an S2S prediction, and removing them is likely to improve the training process, as well as to eliminate spurious predictions resulting from attempts to fit the seasonal cycle or long-term trends. Removing long-term trends and defining MHWs with respect to a changing baseline is also appropriate for shorter-term S2S forecasts and is consistent with a qualitative understanding of an MHW as being extreme in both time and space (Amaya et al., 2023).

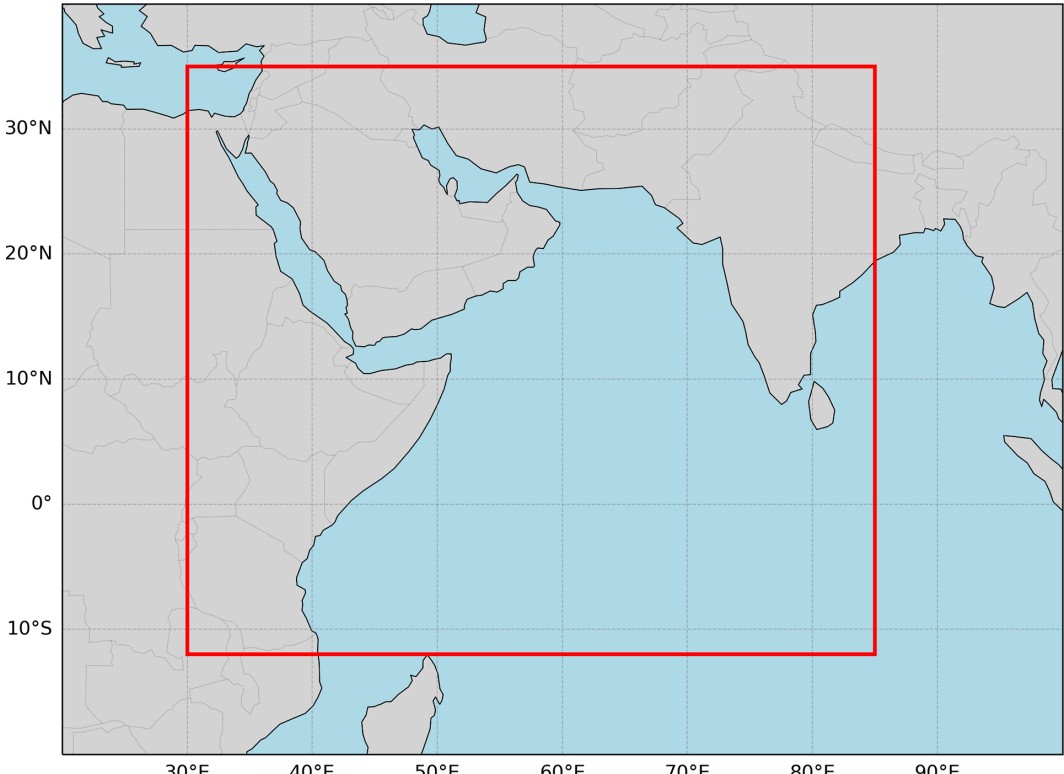

**Figure 1.** *Boundaries of the study domain in the Indian Ocean and Arabian Sea are delineated in red. This represents the region in which ECMWF forecast skill will be evaluated, as well as the domain of the ML forecast model.*

### 2.2. ML model

An initial hyperparameter tuning was performed manually to identify promising parameters. This tuning was relatively minimal, with several choices tested for the number of hidden layers (4, 8, and 16) and the number of feature maps (4, 8, 16, and 32 in the first layer). The resultant models all performed well, and a more rigorous tuning process was therefore not attempted. Three different NN architectures were tested after preliminary hyperparameter tuning in an ablation study, with results detailed in Supplementary Appendix A. The tested architectures were a standard U-Net, a U-Net with skip connections removed, and an NN with only convolutional layers. The main body of the paper will address only the U-Net architecture.

In this work, we utilize NN models to predict SST up to 10 weeks in advance probabilistically. The architecture is shown in Figure 2. The detrended reanalysis data are treated as the true temporal evolution of the system. Rather than using a recurrent architecture (as in Taylor and Feng) to capture temporal dynamics, we provide "true" SST and SSH at the forecast initialization time, as well as at weeks 1–4 before the forecast initialization time, similar to the approach used by Davenport et al. for interannual SST forecasts in which the state at multiple lag times is provided as input for prediction.

The NN output specifies forecasts at each grid point and consists of two channels parameterizing a probability distribution of SST: one channel predicts the mean ($\mu$) and the other the standard deviation ($\sigma$). This can be conceptualized as the summary statistics of an infinite-sized forecast ensemble (Leutbecher and Palmer, 2008). Compared to Xie et al., which incorporates atmospheric variables to predict daily SST at 1–30 days deterministically, our architecture uses only ocean variables as predictors and probabilistically predicts weekly SST at 1–10 weeks.

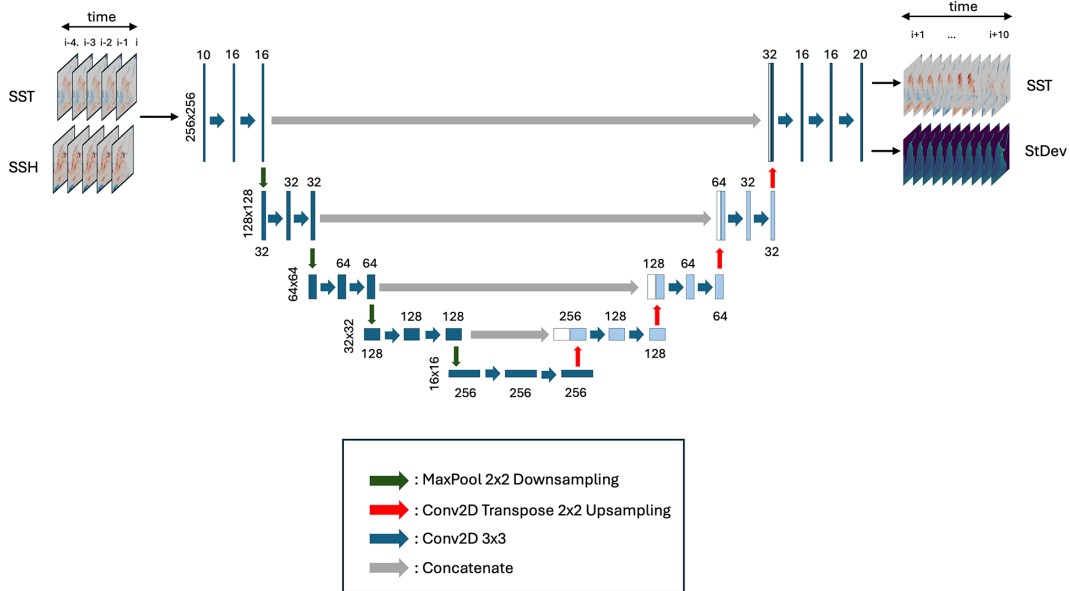

**Figure 2.** *Schematic of U-Net architecture. The input is SST and SSH at the current time step and the four previous time steps. Output is predicted SST and SST uncertainty, represented as standard deviation, at 1–10 weeks lead times. There are four levels of downsampling* via *Maxpooling (green arrows), followed by four levels of upsampling* via *transposed convolution (red arrows). Standard convolutional layers are applied after each up- or downsampling step. After each upsampling layer and before the convolutional layer is applied, data from the downsampling with matching resolution is concatenated (gray arrows).*

This approach differs from traditional ensemble forecasting methods, either dynamical or ML-based, which typically generate multiple realizations to represent uncertainty (Price et al., 2025). Instead, our model directly learns to predict both the expected value and the associated uncertainty from historical patterns in a single forward pass. The uncertainty estimates incorporate both aleatoric uncertainty (inherent variability in the system) and epistemic uncertainty (limitations in the model's predictive capacity), particularly important at longer lead times where predictability decreases.

The Continuous Ranked Probability Score (CRPS), a metric often used for probabilistic forecasts, was employed as the cost function to be minimized during model training (Gneiting and Katzfuss, 2014; Chapman et al., 2022). For our Gaussian parameterization, the CRPS can be analytically computed as:

$$\mathrm{CRPS}(F, y) = \sigma \left[ \frac{1}{\sqrt{\pi}} - 2\phi\left(\frac{y-\mu}{\sigma}\right) - \frac{y-\mu}{\sigma}\left(2\Phi\left(\frac{y-\mu}{\sigma}\right) - 1\right) \right] \tag{2.1}$$

where $F$ represents the predicted cumulative distribution function (CDF) parameterized by $\mu$ and $\sigma$, $y$ is the observed value, $\phi$ is the standard normal probability density function, and $\Phi$ is the standard normal CDF. This cost function simultaneously penalizes forecasts that are either imprecise (large $\sigma$) or inaccurate (large difference between $\mu$ and $y$), encouraging the model to produce calibrated uncertainty estimates. Unlike deterministic loss functions such as mean squared error, CRPS rewards models that correctly quantify prediction uncertainty, which is essential for subseasonal forecasting, where the skill of a deterministic forecast is fundamentally limited. Training was performed using the ADAM optimizer (Kingma and Ba, 2017) with a learning rate of 1e-4. Early stopping was used to reduce the risk of overfitting by monitoring loss on the portion of the dataset held out from training.

## 2.3. *Forecast skill metrics*

The SST forecasts generated by the NN models can be treated as deterministic, and standard metrics used for evaluating deterministic forecasts are applied in this study. We present two metrics: the root mean squared error (RMSE) and the anomaly correlation coefficient (ACC). Deterministic metrics (RMSE and ACC) are applied to the U-Net forecast using the mean SST predictions. Persistence forecasts (in which the current weekly mean SST anomaly immediately before the initialization time is projected to continue indefinitely) and a forecast based on climatology (in which the forecasted anomaly is zero for all times) will be used to evaluate the skill of the U-Net.

To evaluate the full probabilistic forecast, the output from the NN must first be converted from an SST predictor to an MHW predictor. To do this, a spatially varying threshold equal to the 90th percentile of weekly mean SST was used to define an MHW event. When using daily SST, an MHW is usually defined by a threshold and minimum duration (Hobday et al., 2016). For longer timeframes in which monthly SST data are more appropriate, the 90th percentile threshold is used with no minimum duration. Here we follow the latter approach, applied to the weekly mean SST, as the typical minimum duration (5 days) is shorter than 1 week.

After computing the MHW threshold, the NN forecast is treated as a normal distribution, and the exceedance probability can be calculated as:

$$P_{MHW} = 1 - F_{\mu,\sigma}(T_{th}). \tag{2.2}$$

where $P_{MHW}$ is the predicted probability of an MHW event, $F_{\mu,\sigma}$ is the normal CDF with U-Net-predicted temperature $\mu$ and standard deviation $\sigma$, and $T_{th}$ is the MHW SST threshold value.

We then compute the Brier Skill Score (BSS) (Gneiting and Katzfuss, 2014) and Symmetric Extremal Dependence Index (SEDI) (Ferro and Stephenson, 2011). The Brier Score is a generalization of RMSE to probabilistic forecasts, where the error is equal to the difference between the predicted probability of an event and its actual occurrence (which is binary). BSS is defined as

$$BSS = 1 - \frac{BS}{BS_{ref}}, \tag{2.3}$$

where $BS$ is the Brier Score and $BS_{ref}$ is the Brier Score of a reference forecast. BSS has a maximum possible value of 1, with values greater than zero representing skill compared to the reference forecast. Here, we use an MHW forecast probability of 0.1 for all times and locations as the reference. This is consistent with the MHW definition we applied and corresponds to a background MHW risk of 10%. Confidence intervals were estimated following Bradley et al. (2008).

SEDI is a metric used to assess the skill of probabilistic predictions for rare binary events and is commonly used to evaluate MHW forecast tools (Jacox et al., 2022; Mogen et al., 2023). The significance threshold was computed following the method employed by Jacox et al. Random sequences of 10 weeks of SST data were taken as climatology-based forecasts, with 1,000 random forecasts generated. The SEDI score was calculated for each forecast, and the 97.5th percentile of these scores was set as the significance threshold. For the purpose of computing SEDI, the U-Net exceedance probabilities were converted to binary MHW forecasts by using a threshold of 50%, with forecasted probabilities >50% corresponding to a forecasted MHW and probabilities <50% corresponding to no forecasted MHW.

After training on the first 80% of the time series (January 3, 1993–October 25, 2015), the U-Net forecast model was applied to the remaining 20% for testing (November 1, 2015–June 30, 2021) with no separate validation split used for hyperparameter tuning. While this does potentially limit generalization, it is sufficient as a proof-of-concept as opposed to operational implementation and is consistent with some other AI/ML ocean forecasting applications (Liu et al., 2021; Tian et al., 2022). All metrics were calculated on this test dataset, and all results presented averaged over space represent area averages of skill rather than skill of area-averaged forecasts. The performance of the U-Net on the training dataset is not included, as it is not indicative of generalization to out-of-sample data not represented by the training data.

### 2.3.1. ECMWF baseline

The best-performing NN model will be compared to the performance of a dynamical S2S model to assess its performance compared to traditional modeling tools. The European Centre for Medium-Range Weather Forecasting (ECMWF) generates an ensemble S2S predictive model that forecasts daily mean SST at lead times of 1–46 days. The model is an ensemble forecast with one control member and 50 perturbed realizations (Vitart et al., 2017). Forecasts initialized during the test period will be processed into forecasts of weekly mean SST covering the same geographical domain. The probability of an MHW event will be calculated, assuming the SST distribution at a single location is normal, and using the sample standard deviation of the ensemble to compute the CDF. Alternatively, the discrete ensemble can also be used to calculate an empirical CDF, but the results were similar using either method.

## 3. Results

Before moving to showing the performance of the fully trained model on the test dataset, we first show the training results in Figure 3. This plot shows the performance of the model during training and is the first step in ensuring that the trained model is not overfit. CRPS loss (Figure 3a) and RMSE (Figure 3b) are both included. The model's performance on the held-out test dataset is monitored during training to prevent overfitting. With test loss and RMSE both plateauing before reaching the maximum number of epochs, training was automatically terminated, despite the training loss and training RMSE continuing to decrease. The flat test loss at the end of training and the small performance gap between training and test metrics are good indicators that no overfitting has occurred.

Following training, the predictive performance of the U-Net, including its SST and MHW predictions on the test dataset, was evaluated using the metrics specified in Section 2.3. Ideally, the forecasts produced will be more accurate compared to climatology and persistence references. Additionally, the U-Net should be able to predict the probability of MHW occurrence with more skill than an estimate based on background incidence and produce physically plausible forecasts of SST anomalies.

The forecast metrics (deterministic and probabilistic) as a function of lead time, averaged over all time and space, are shown in Figure 4. The U-Net outperforms both climatology and persistence at all lead times by these metrics. Persistence exceeds climatology RMSE at week 3 (Figure 4a), with the U-Net converging to climatology by week 10. The ECMWF forecast has an RMSE larger than the U-NET for all

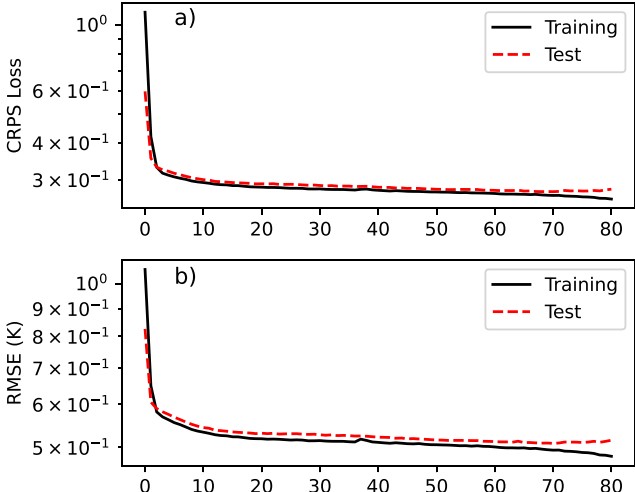

**Figure 3.** *Model performance during training. Panel (a) shows the CRPS loss, and panel (b) shows the RMSE, each plotted for both the training (black) and test (red) datasets. Decreasing values indicate improving performance, and the convergence of the curves suggests limited overfitting.*

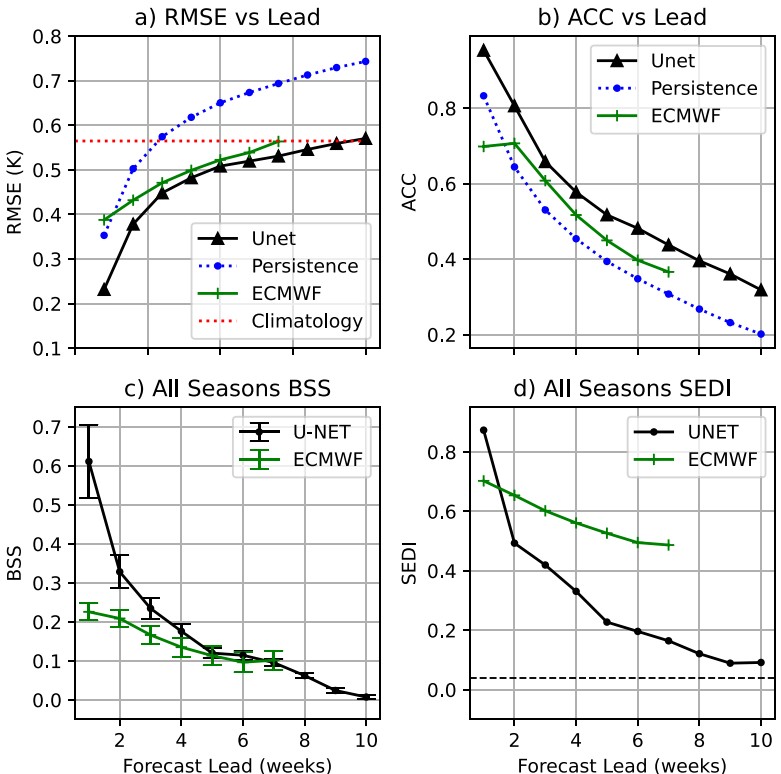

**Figure 4.** *Forecast skill metrics for the U-Net as a function of lead time. Panels (a) and (b) show deterministic metrics: RMSE (a) and ACC (b). For comparison, both include a persistence forecast, and the climatology RMSE is also shown in (a). Panel (c) shows the BSS averaged over space and time, and panel (d) shows the SEDI. Ninety-five percent confidence intervals are included for all metrics, and the SEDI significance threshold is shown in (d) as a dashed black line. Metrics in panels (a–c) are based on SST predictions, while panel (d) uses binary MHW forecasts.*

lead times, converging to climatology at the end of its forecast at week 7. The ACC of a persistence forecast drops below 0.5, below which forecasts are generally not considered skillful, at week 3. At the same time, the U-Net does not fall below this threshold until week 6, and the ECMWF forecast at week 5 (Figure 4b).

The all-season BSS (Figure 4c) shows statistically significant skill for the U-Net forecast at all 10 weeks of forecast lead time. ECMWF forecasts also have significant skill for all lead times, with similar skill at weeks 6–7 and notably worse skill for shorter lead times. In contrast, ECMWF has notably better SEDI scores than the U-NET for most lead times. The U-Net forecast has statistically significant skill for most lead times, before dropping below the significance threshold at week 8. In contrast, ECMWF maintains significant skill for all forecasted lead times (Figure 4d).

When broken down by season (Figure 5), using BSS, the U-Net shows statistically significant skill in all 10 weeks of lead time across all seasons. Using SEDI, boreal summer has statistically significant skill out to 7 weeks, with other seasons generally retaining skill for longer. Boreal fall does drop slightly below the significance threshold at week 6 before regaining skill in later weeks; this is likely due to the noise associated with transforming predictions of continuous probability distributions into predictions of discrete events.

In addition to having demonstrable skill in forecast metrics, a probabilistic forecast should be reliable (i.e., events occur with frequencies consistent with its predictions) and have useful resolution (i.e., it

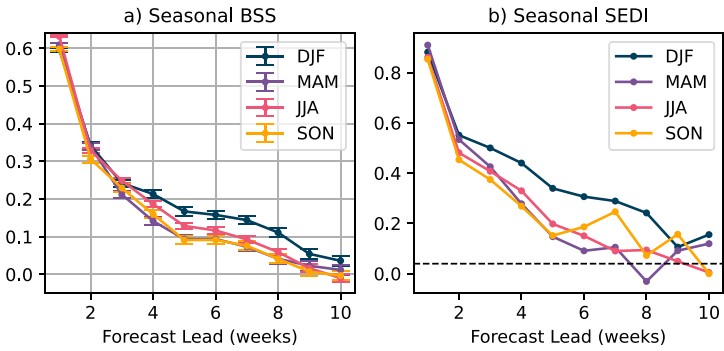

**Figure 5.** *BSS (a) and SEDI (b) for the U-Net, shown by season as a function of lead time. Seasons are defined as: December–February (DJF, boreal winter), March–May (MAM, boreal spring), June–August (JJA, boreal summer), and September–November (SON, boreal fall). For panel (a), 95% confidence intervals are shown for each lead time. For panel (b), the significance threshold is indicated by a dashed black line. BSS is computed from SST forecasts, while SEDI is computed from binary MHW forecasts.*

should predict a relatively wide range of probabilities for different events). We use a calibration curve to evaluate the U-Net's reliability and resolution, as shown in Figure 6.

For the first 3–4 weeks, the U-Net is very well calibrated with the actual and predicted probabilities closely following the ideal $y = x$ curve. The noticeable flattening of the curve at weeks 7 and above indicates that the U-Net can no longer discriminate between different levels of elevated MHW risk, with actual incidence well below the frequency predicted by the U-Net. However, even at week 10, with a predicted probability of 0.2–0.3 (representing 2–3 times the background MHW risk), the U-Net produces a reliable probabilistic forecast. The U-Net calibration compares favorably with the ECMWF forecast, which generally systematically underpredicts the probability of an MHW event across a range of probabilities for all lead times.

Finally, the spatial distribution of the probabilistic skill metrics (BSS and SEDI) is shown. Unlike Figure 4, which shows the metrics computed across all time and space, Figure 7 displays BSS averaged only over time. Panel (a) shows the spatial distribution of BSS for the ECMWF forecast. Panel (b) shows the spatial distribution of BSS for the U-Net forecast. In both plots, values greater than zero indicate skill (green) and values less than zero indicate a lack of skill (blue) compared to a reference climatology forecast. Both models show similar patterns, with the skill degrading most rapidly near the African coast. Both ECMWF and the U-Net retain BSS skill in most of the domain for all forecast lead times. The warm pool, straddling the equator, retains the most skill for the longest, with skill lost in most of the Red Sea by week 5.

The spatial distributions of SEDI scores for the ECMWF and U-Net forecasts are also included for comparison (Figure 8). SEDI exhibits similar spatial patterns to BSS, with the warm pool demonstrating the greatest skill and retaining it for the longest. Unlike BSS, the U-Net forecast has lost skill over most of the domain by week 10, with the Red Sea losing all skill by week 3. Skill is retained for the longest in the warm pool, as in BSS. This pattern, in which areas of more rapid skill degradation correspond to the warm pool region, is consistent for both models and metrics, even as the U-Net's SEDI score drops more rapidly with increasing lead time compared to ECMWF.

## 4. Discussion

While ACC and RMSE are similar metrics and are often employed for similar purposes, they measure different aspects of forecast quality. For the U-Net, they tell consistent but different stories. Using the commonly applied threshold of ACC = 0.5 to delineate skillful/not skillful forecasts, the U-Net loses skill

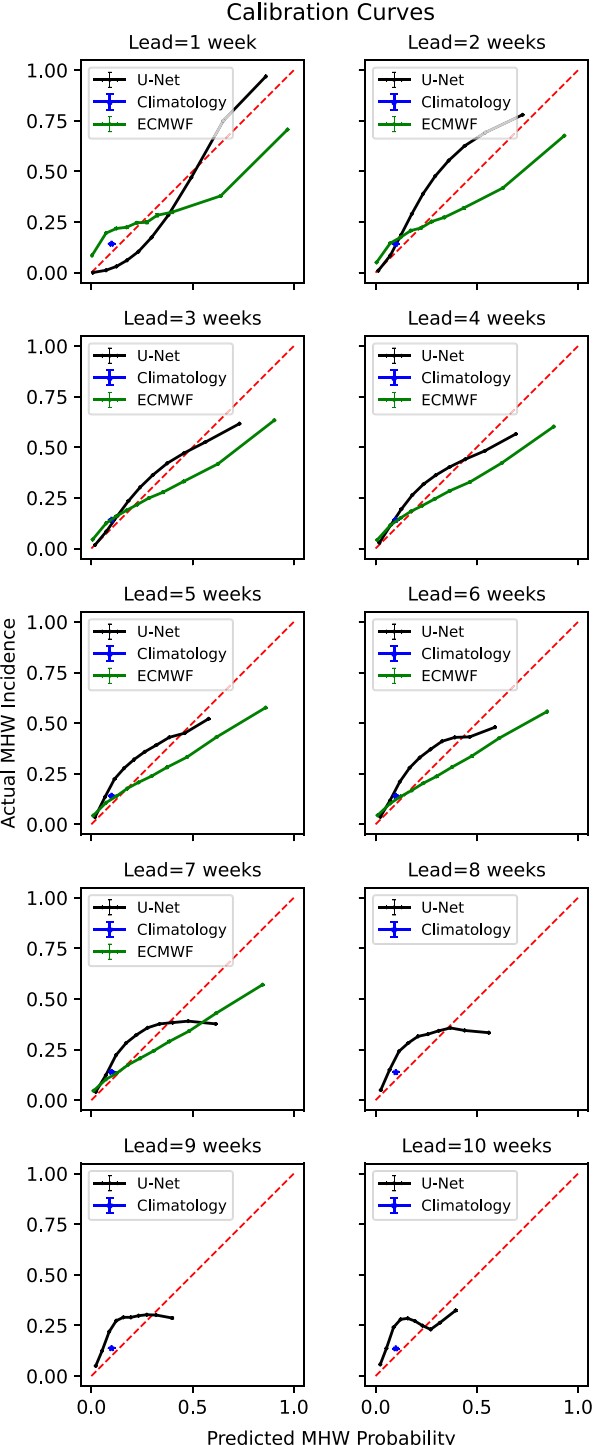

**Figure 6.** *Calibration curves for all lead times of U-Net and ECMWF forecasts. For each lead time, predicted probabilities are binned along the x-axis. The actual incidence of these events is then shown on the y-axis. A perfectly calibrated forecast, where events occur at exactly the predicted frequency, is included for reference. A climatology-based prediction is also included in blue for all lead times. 95% confidence intervals are included in both the x and y directions for each point. Curves below the 1:1 line indicate overprediction, while curves above indicate under-prediction.*

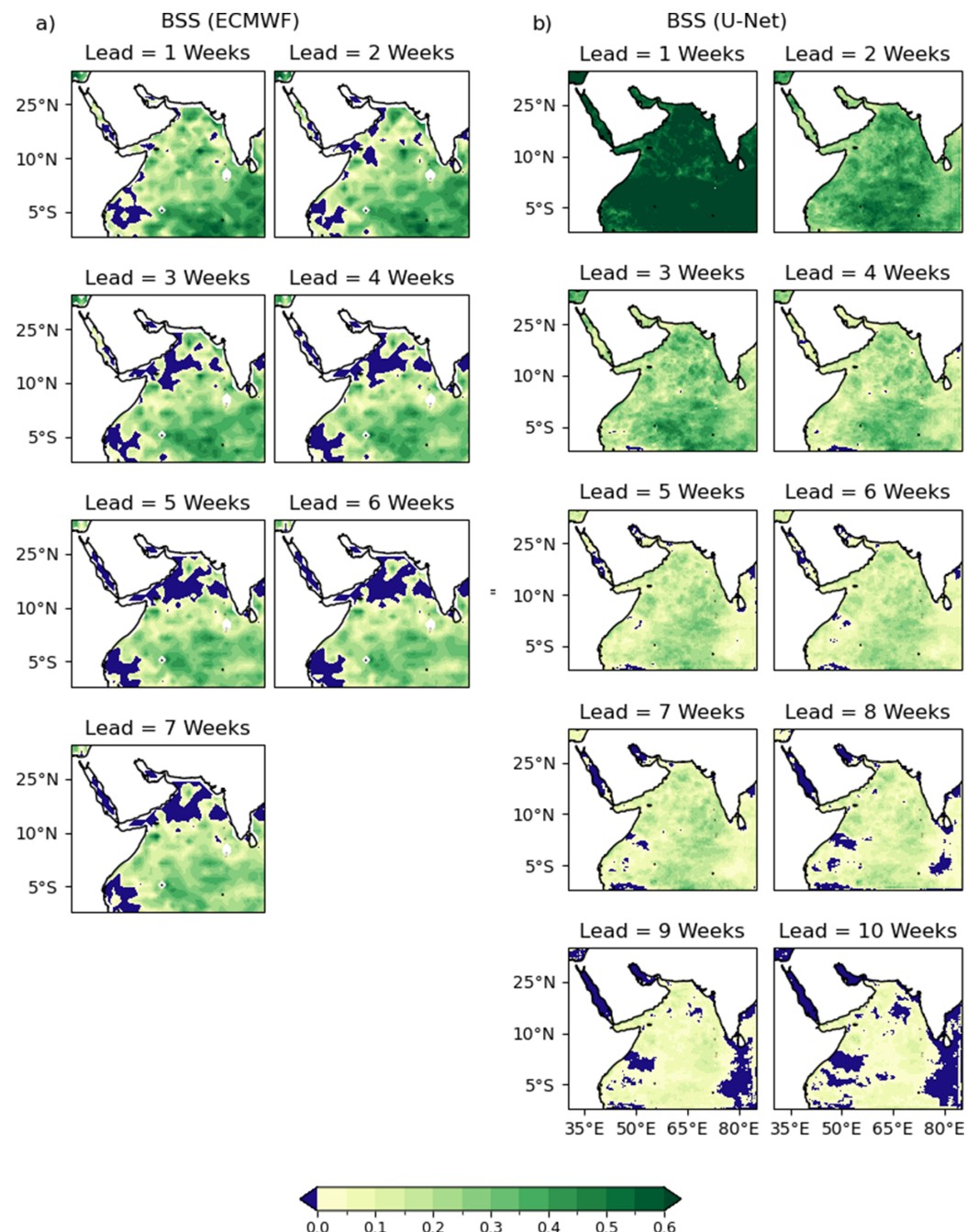

***Figure 7.*** *Maps showing the spatial distribution of the Brier Skill Score for the ECMWF forecast (a) and U-Net forecast (b). Areas with skill scores less than zero are shown in blue, with green shades representing skillful forecasts compared to climatology.*

after week 4. ACC values <0.5 generally correspond to RMSE values that exceed the climatology. However, here, the RMSE remains below the climatological RMSE for all lead times. This is consistent with the expected behavior of statistical models, which revert to climatology with longer lead times, thereby reducing RMSE even as the correlation drops.

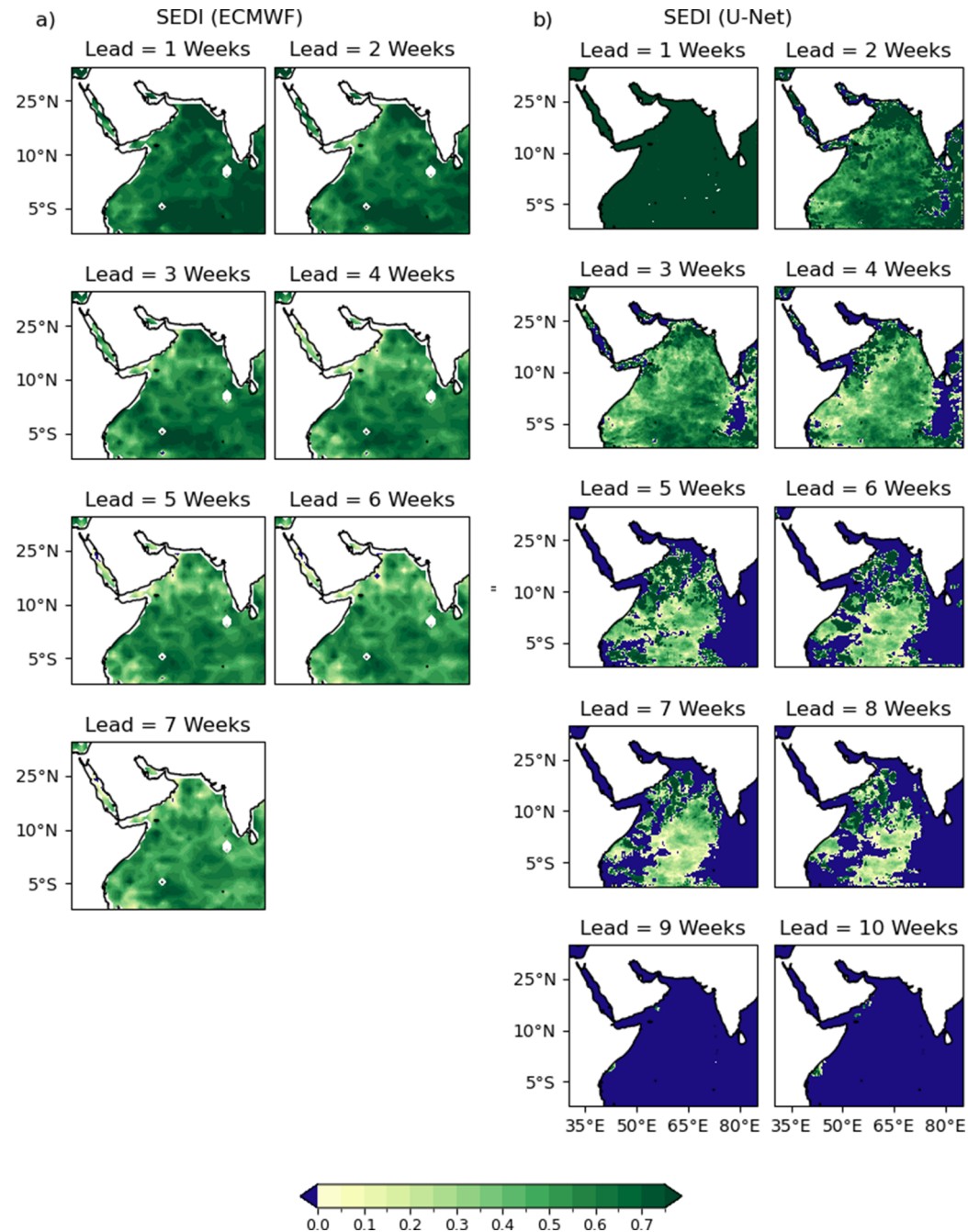

**Figure 8.** *Spatial distribution of the Symmetric Extremal Dependence Index for the ECMWF forecast (a) and U-Net forecast (b). Areas with negative skill (SEDI < 0) are shown in blue. For the U-Net, a binary forecast was generated by applying a 50% threshold to the predicted MHW incidence probability (equation 2.2). Positive SEDI values indicate skillful detection of extreme events relative to climatology.*

This is explained by the statistical, rather than dynamical, nature of the U-Net. Rather than generating physically plausible individual trajectories of SST, the U-Net is trained to generate statistically reliable and precise projections of SST at each of its lead times. As a result, for longer lead times, the U-Net

prediction slowly reverts to a forecast of climatological mean and standard deviation. RMSE will exceed climatology for ACC < 0.5 if the forecast variance is equal to the true variance. Decreasing the forecast variance by reverting to climatology, as the U-Net does, allows it to retain useful RMSE skill even as the ACC drops below 0.5.

Existing MHW forecast tools (both dynamical and statistical) also show this effect, particularly those that make forecasts on seasonal time scales. Ensemble forecasts using dynamical models will trend toward climatology at longer lead times as the ensemble members become decorrelated. Jacox et al. (2022), using an ensemble-based forecast, show skill-approaching climatology at the actual predictability limit of the model.

The statistical calibration presented in Figure 6 shows a general pattern of under-predicting low-risk and overpredicting high-risk events. This pattern becomes more pronounced for longer lead times. This is likely due to the relative rarity of MHW events (comprising 10% of the training data by definition). A more extensive training dataset, using higher resolution data, a larger geographic region, or a longer time period could mitigate this behavior.

The spatial distribution of skill presented in Figures 7 and 8 show that the region of the Indian Ocean straddling the equator away from the coasts retains the most skill for the longest. This pattern is evident for both BSS and SEDI. The spatial distribution of RMSE and ACC skill shows a similar pattern. This area roughly corresponds to the Indo-Pacific warm pool, where warm water extends deeper year-round than in other regions. This depth of warm water will tend to have a longer memory and less variability, expanding the horizon of predictability—consistent with the skill of the U-Net. It is likely that SSH, which is one of the input variables provided to the U-Net, is able to act as a proxy for this source of predictability, as ocean heat content will impact SSH and be usable implicitly by the model for predictions.

The comparison with ECMWF skill for BSS and SEDI appears contradictory, as the U-Net has significantly larger BSS skills for short lead times and comparable scores by the end of the ECMWF forecast period. In contrast, except for week 1 forecasts, the U-Net has much worse SEDI scores for all lead times. This can be explained by the different aspects of probabilistic forecast skill that each metric is designed to capture. SEDI only evaluates a binary forecast and only rewards a true positive if the predicted probability is >50%. ECMWF S2S forecasts are known to be underdispersive (Mayer et al., 2024), resulting in a more sharply peaked distribution and a larger proportion of forecasts in which the predicted MHW exceedance probability exceeds 50%. For longer lead times, the U-Net rarely makes predictions that reach that level of confidence, even when it correctly identifies elevated MHW risks. The latter are taken into account in BSS, but not in SEDI, which explains the discrepancy between Figures 4c and 4d.

The contrasting spatial patterns observed in Figures 7 and 8 provide further insight into the fundamental differences between the U-Net and ECMWF forecasts. In Figures 7b and 8b, after week 7, the green areas (indicating skill) in the U-Net results become unevenly distributed, with blue areas (indicating lack of skill) occupying larger proportions. In contrast, Figures 7a and 8a show that the green distribution remains more uniform from week 1 to week 7 in the ECMWF model. This difference likely stems from the data-driven versus dynamical approaches employed. The U-Net, while it can capture spatial autocorrelation via training, has no explicit flow dynamics built in and may produce forecasts that display relatively high levels of independence between grid points. In contrast, the ECMWF model, being physics-based, naturally and explicitly maintains flow dependence between adjacent grid points. This flow dependence in the ECMWF model helps maintain spatial coherence in skill patterns, whereas the U-Net may exhibit more heterogeneous and less spatially autocorrelated skill patterns at longer lead times.

Given this discrepancy between the skill of the U-Net and ECMWF models on BSS and SEDI skill, it is important to consider the use case and purpose of the forecast when building and evaluating forecast models of MHW events. Targeting a forecast that is statistically well calibrated, as we did here, may result in forecasts that are relatively conservative for extreme events with confidence that does not exceed a 50% threshold, even for strong signals. Stakeholders may find a model that makes binary predictions, although it is less well-calibrated, to be more useful in certain situations. For ML models, this can be addressed by choosing an appropriate loss function. Extensions of this work may include retraining using alternative loss functions to make the ML-generated MHW forecasts more fit-to-purpose for stakeholders. Using

SEDI or other similar metrics as a cost function could potentially result in a trained model that produces more useful binary predictions than the model presented in this work.

Future work may also include using explainable artificial intelligence (XAI) techniques to identify the sources of predictability for specific events forecasted by the U-Net. While dynamical S2S MHW forecast tools have meaningful skill, identifying the primary physical phenomenon driving that skill for particular events is often challenging. XAI techniques applied to the U-Net could provide insight into the sources of predictability, identify forecasts of opportunity, and inform additional studies of dynamical forecast tools.

## 5. Conclusion

Marine heatwaves can significantly impact ecosystems and human populations, and reliable forecasts of these events on S2S timescales are highly valuable. MHWs throughout the Indian Ocean, and specifically in the Arabian Sea, are particularly interesting due to the large population densities adjacent to these bodies of water, their economic importance, and the potential coupling of MHWs to atmospheric phenomena. While dynamical models and ML forecast tools have both demonstrated skill on S2S timescales, ML tools have generally been deterministic. Regional models, on the other hand, have not focused on the Arabian Sea or the Indian Ocean, and predictions have been made on monthly timescales instead of weekly ones. Here, we present a regional ML MHW forecast tool producing skillful probabilistic predictions out to 10 weeks of lead time.

The forecast produced by the trained ML model produces skillful predictions compared to persistence and climatology-based estimates across a range of deterministic and probabilistic metrics. RMSE is lower compared to climatology out to 10 weeks and outperforms persistence at all lead times. BSS and SEDI are statistically significantly greater than zero for all lead times. Broken down seasonally, the model retains skill for summer events at 6 weeks, while all other seasons retain skill for longer lead times. The model is generally well-calibrated; the frequency of events is consistent with the predicted probabilities, with performance degrading as expected for longer lead times. Forecast skill is comparable to or better than ECMWF S2S forecasts across most metrics, and it extends statistically significant skill out to 10 weeks, compared to a forecast horizon of 7 weeks generated by ECMWF's model.

Spatially, skill is retained longest in the region of the warm pool. This is consistent with expectations based on physics (a deeper warm layer will have less variability and more predictability) and prior work showing that these regions retain MHW forecast skill for longer than others.

Further refinement of the ML architecture or the use of additional training data could yield improvements in U-Net performance. It is worth exploring the application of generative AI methods, such as diffusion or the use of recurrent architectures. Applying XAI techniques to the U-Net as-is could also yield insights into the physical sources of predictability and contribute to improvements in dynamical S2S models. Finally, the skillful SST predictions demonstrated in this paper can potentially improve atmospheric S2S forecasts in coupled models.

**Open peer review.** To view the open peer review materials for this article, please visit http://doi.org/10.1017/eds.2026.10033.

**Supplementary material.** The supplementary material for this article can be found at http://doi.org/10.1017/eds.2026.10033.

**Author contribution.** LH performed experiments, including training of the ML model. The study was conceived by LH and ACS, with input on experiments from JRN, DG, and IH. The manuscript was drafted by LH, with revisions by ACS, JRN, DG, and IH. ACS, DG, and IH acquired funding for the project. ACS provided supervision.

**Competing interests.** The authors declare none.

**Data availability statement.** Code and processed data used for generating the results in this manuscript are publicly available Howard, 2024. Raw reanalysis data used for training and dynamical S2S forecast data are freely available online from their originating agencies at https://data.marine.copernicus.eu/product/GLOBAL_MULTIYEAR_PHY_001_030/description.

**Funding statement.** Support for this work was provided by KAUST CRG Grant ORA-2021-CRG10–4649.2. Giglio also received support from the NASA award 80NSSC21K0556. This work utilized the Alpine high-performance computing resource at the University of Colorado Boulder. Alpine is jointly funded by the University of Colorado Boulder, the University of Colorado

Anschutz, Colorado State University, and the National Science Foundation (award 2,201,538). This work additionally utilized the Blanca condo computing resource at the University of Colorado Boulder. Blanca is jointly funded by computing users and the University of Colorado Boulder.

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
