## [Reviewer Report]

1.Should “Marine Heat Waves” be abbreviated as MHW or MHWs? There are multiple instances in the text where the abbreviation is used both with and without the plural “s”; consistency should be maintained.

2.“With 256 grid points in both the latitudinal and longitudinal directions.” While this is convenient for calculations, differences in the span of latitude and longitude result in variations in the grid distances along the latitudinal and longitudinal directions.

3.In “2.2 Machine Learning Model”, should the domestic and international application cases of U-Net be placed in the introduction?

4.On Page 4, line 40: “we provide SST and SSH at the forecast initialization time as well as at weeks 1-4 before the forecast initialization time, similar to the approach used by Davenport et al for interannual SST forecasts.” The source of SST and SSH at the forecast initialization time is not clearly stated.

5.In Figure 4, please provide the original expansions of the abbreviations such as DJF, MAM, JJA, and SON.

6.From Figure 6, it can be seen that after week 7, the green areas in the U-Net results are unevenly distributed, with blue areas occupying a large proportion. In contrast, in Figure 7, the green distribution is very uniform from week 1 to week 7. Should it also be analyzed that the training method of the U-Net leads to relative independence between grid points, while the ECMWF model has good flow dependence?

---

## [Reviewer Report]

This paper describes a data-driven subseasonal forecasting systems for sea surface temperature in the North Indian Ocean and the Arabian Sea, used to forecast marine heatwaves. Sea surface temperature and sea surface height from a high-resolution global reanalysis are used to train a U-Net neural network architecture. The resulting forecasts show significant deterministic skill of SST and MHW occurrence several weeks ahead across the domain, with exceptions noted. The proposed system outperforms benchmark models and compares well with a state-of-the-art dynamical system.

The paper is well written and largely logical, and the figures are clear. A suitable range of validation techniques are used to explores geographical patterns and seasonality of skill. The development of data-driven models, particularly for this area, is justified and of worth to the subseasonal forecasting community.

There are some major concerns to be addressed. I am particularly concerned about the probabilistic nature of the forecast system, as the method used to create the ensemble is not well explained in the manuscript and no adequate references.

Major Comments

Probabilistic forecasting. To my knowledge (which may be lacking), a true data-driven forecast ensemble has only been produced for short-term forecasting (Price et al., 2025). A much more detailed explanation is required here on the method used to generate the ensemble. What does this probability distribution of SST represent? How is it calculated? How does the UNet output this? How large is it? This information is crucial to any paper on S2S forecasting.

Price, I., Sanchez-Gonzalez, A., Alet, F. et al. Probabilistic weather forecasting with machine learning. Nature 637, 84–90 (2025). https://doi.org/10.1038/s41586-024-08252-9

More importantly, how does it compare to a dynamical ensemble, which represents uncertainty in initial conditions and chaotic nature of the evolution of the earth system (particularly for atmospheric and sea surface variables on S2S timescales)? If the nature or the size of the ensembles of the data-driven and dynamical models are different, then comparisons are not fair and the skill scores potentially used incorrectly.

Describe the persistence in some more detail. What does “current SST anomaly” refer to – daily, weekly, before start date?

The beginning of the results section is hard to follow as the training, validation and test have not been clearly defined. Moreover, I find the captions could do with more explicit information. Please guide the reader better.

Since the first map does not appear until Figure 6, consider recording the figures or adding a description and definition of the domain.

In the discussion, the authors state ideas for potential improvements (Pg 12, Lines 47-51). The authors should state why they think these are useful next steps. The authors reduced the resolution of the reanalysis training data used; why not show from the beginning the skill of using the original dataset (1/12)?

Please explain the hyperparameter tuning so that it can be reproduced.

Justify choice of training data. Why use a reanalysis for SST and SSH for training over the 1993-2021 period when this is covered by observational data?

Why did the authors regrid the reanalysis data?

I strongly recommend to specify the target domain in the title.

Introduction

Line 4: MHWs are not just mere anomalies.

Line 34: Years missing from references.

Line 48: sources

Line 48: sources on what timescales?

Methods

Pg 3

Line 19: 8km at the equator?

Line 26: This is an unusual way to describe the new resolution. Can you be explicit?

Line 29: It is unclear what you mean by “power of two”.

Line 30: Is the detrending performed individually for each week of the year?

Line 39: One could argue that the “original definition” of MHWs is the fixed climatology used in Hobday 2016. I would recommended to change wording.

Line 42: Introduce NN acronym here.

Line 33: Why were these three architectures chosen? What do they represent?

Line 43: How is the approach similar to Davenport et al?

Pg 5

Line 47: Please explicitly state the training and testing periods. Also, isn’t there also a validation period?

Pg 6

Lines 16-21. This paragraph seems to have lost some sentences and it not clear.

Line 35: AT the end of section 2.3, you state that the training results are not include but here you discuss them. Please clarify.

Line 43: “more accurate”.

Pg 7

Line 47: Is there a more precise term than “proper”?

Pg 9

Lines 40-44: I disagree that skill degradation is consistent. There is much more skill degradation in the U-Net SEDI.

Pg 12

Line 45. Do you mean Figure 5?

Line 48: What becomes more extreme? Unclear.

Pg 13

Line 6: Represented in the SSH, perhaps?

Line 23: Would be good to mention in the methods already that this is a binary forecast.

Pg 14

Line 4: Please be realistic here, as sometimes the skill is worse than the dynamical system e.g. SEDI scores.

Figures

Figure 2: Here, and in general, are we looking at area-averaged skill or skill of area-averages?

Figure 3: Specify if the top panels are for SST and the bottom panels are for MHWs.

Figures 6 & 7: To allow for easier comparison, please considering putting the skill scores for the two systems side-by-side. Also, there is a colorbar missing.

---

## [Editor Report]

Following reviewers' comments, I recommend major revisions. Nevertheless, please note that the revisions needed are mostly clarifications and elaborations. Especially, regarding the probabilistic prediction, even if the approach is described in referenced papers, it is relevant to give more details on the application for your particular case.

---

## [Reviewer Report]

All the relevant issues raised in my first-round review have been adopted and revised by the authors. I recommend acceptance for publication.

---

## [Editor Report]

Thank you for submitting your revised version. As you can see, only one reviewer provided feedback. I have reviewed your responses and the modifications you made to address Reviewer 2’s comments. Based on these, I recommend minor revisions for this version. Please see my comments below (based on Reviewer 2’s assessment):

Use of the term “downscaling”: The term is misleading. Downscaling

generally refers to increasing resolution, which is the opposite of

downsampling. In the manuscript, the two terms seem to be used as

synonyms. Could you please correct this thoughout the manuscript?

P3L29:

“Reanalysis was chosen rather than a data product constructed from observations alone as reanalyses are dynamically constrained and coupled to processes that act as sources of S2S predictability.”

I don’t fully understand this point. Observations represent the real system, which is also dynamically constrained and coupled by nature and contains the true sources of S2S predictability. Do you mean that biases, noise, or insufficient resolution in observations make reanalysis more relevant? Or is there a reference indicating that reanalyses are more accurate than observations?

P4L38: Hyperparameter tuning is not detailed here. You might consider moving the explanation currently in P7L36–37 to this section.

P4L40–43: Your reasoning for using three architectures is very assertive. In theory, all architectures can represent both large and small scales. In practice, differences may exist, but given the small differences between the three architectures, could you consider moving this to the appendix (as an ablation study) and focusing only on UNET in the main paper?

Table 1: Is it referenced in the text? Also, the term “validation” is used, but you acknowledged in your response to Reviewer 2 that you did not have a validation set.

---

## [Editor Report]

Thanks for addressing the last minor comments. I recommend the article for publication in EDS. Congratulations! One minor thing I would suggest changing: in the caption of figure 3 (the embedded caption in the figure), the term validation is still indicated, while it would be more consistent to put “Test”